# Graph vs. Sequence: An Empirical Study on Knowledge Forms for Knowledge-Grounded Dialogue

**Yizhe Yang,  Heyan Huang,  Yuhang Liu,  Yang Gao**[*]

School of Computer Science and Technology, Beijing Institute of Technology
Beijing Engineering Research Center of High Volume Language Information Processing
and Cloud Computing Applications
Beijing Institute of Technology Southeast Academy of Information Technology
{yizheyang,hhy63,codelyh,gyang}@bit.edu.cn

## Abstract

Knowledge-grounded dialogue is a task of generating an informative response based on both the dialogue history and external knowledge source. In general, there are two forms of knowledge: manually annotated knowledge graphs and knowledge text from website. From various evaluation viewpoints, each type of knowledge has advantages and downsides. To further distinguish the principles and determinants from the intricate factors, we conduct a thorough experiment and study on the task to answer three essential questions. The questions involve the choice of appropriate knowledge form, the degree of mutual effects between knowledge and the model selection, and the few-shot performance of knowledge. Supported by statistical shreds of evidence, we offer conclusive solutions and sensible suggestions for directions and standards of future research.

## 1  Introduction

Consistent human knowledge is compulsory to achieve an intelligent AI, except for a massive amount of data provided, whether a well-formed structural knowledge  (Moon et al., 2019) or human instruction  (Ouyang et al., 2022). However, many dialogue systems are not proficient in integrating knowledge and cannot engage humans in a thorough discussion about particular subjects. In order to incorporate knowledge into the conversation, knowledge-grounded dialogue systems have gained popularity (Li et al., 2018; Dinan et al., 2019; Moon et al., 2019; Zhou et al., 2018). The goal is to generate informative and meaningful responses based on dialogue history and external knowledge. So far, researchers have gathered datasets with various knowledge forms, such as the recommendation dialogues with knowledge graph (Moon et al., 2019;

---

[*]Corresponding author

Li et al., 2018) and the open-domain dialogues with Wikipedia (Dinan et al., 2019; Zhou et al., 2018).

Generally, there are two forms of knowledge: knowledge graphs and knowledge text. Knowledge graphs are annotated by human experts and saved as triples like <*head entity, relationship, tail entity*>. Because the knowledge is structured, the system can retrieve and use it to facilitate reasoning (Moon et al., 2019). However, a high-quality knowledge graph is quite labor expensive. Instead, the text naturally serves as a medium of knowledge. knowledge text is derived from knowledge-intensive texts and easily parameterized by large language models. Textual knowledge is unstructured, easy to collect, yet challenging to reason and explain directly.

Different forms of knowledge tend to serve different knowledge-intensive language tasks (KILT) (Petroni et al., 2021; Yu et al., 2022; Mialon et al., 2023; Chowdhury et al., 2023; Pan et al., 2023). Specifically, knowledge-grounded dialogue systems can receive either forms of knowledge graph or knowledge text to facilitate response generation; nevertheless, their performances are quite different regarding response quality and factual consistency, etc. The reason is closely tied to knowledge's characteristics. For instance, OpenDialKG  (Moon et al., 2019) is a typical knowledge graph dataset, which annotates the reference triples based on a large knowledge graph for recommended conversations. However, the knowledge in OpenDialKG comes from Freebase (Bast et al., 2014), which requires large-scale data annotation, and the knowledge is sparse (1-2 triples) for generation. On the contrary, WoW dataset (Dinan et al., 2019) retrieves Wikipedia passages for open-domain dialogue to assist intelligent agents in generating informative utterances. The information retrieved in WoW is too dense (i.e., 7 passages) and noisy for models to figure out the accurate knowledge. Thus, an extra selection module is re-

quired to polish the pertinent knowledge. (Zhao et al., 2020b). Therefore, determining the forms of knowledge is crucial and merits extensive discussion in light of the task's features.

With a particular form of knowledge in hand, we have to choose an appropriate approach to incorporate the knowledge to support relevant tasks. Though knowledge-grounded tasks are extensively studied, the majority aiming for advanced performance, few of them have exhaustively compared the determinants in the model and knowledge. Prabhumoye et al. (2021) proposed a large-size Dual-Encoders model on WoW. However, according to our experiments, the base-size Dual-Encoders model may perform better on WoW data. Li et al. (2022) experimented with an Encoder-Decoder model on multiple datasets but ignored the Dual-Encoders are more compatible with the task. Dziri et al. (2022a) experimented on sentence-level knowledge datasets, neglecting the effect of knowledge size and granularity. Nevertheless, comparisons from multiple perspectives are needed to support the superiority and contribution of the model theoretically. However, considering all factors every time in each specific research is not feasible. Therefore, to aid future research, we comprehensively analyze the influencing factors, such as model architecture, model size, pretrained models, etc., and then summarize several determinant conclusions. Concretely, we leverage the scientific strategy of controlling variables to investigate how intricate factors affect and to what extent the model and knowledge should be mutually adaptive.

Furthermore, we also explore how various knowledge performs in few-shot settings as few-shot learning with the knowledge-auxiliary is an important application (Chen et al., 2021). To answer the above question comprehensively, we employ a unified Transformer (Vaswani et al., 2017) framework to observe the effects of different knowledge and serialize the knowledge graph to adapt to the input. In this paper, our investigation mainly focuses on three points:

1. Graph v.s. Sequence, which form of knowledge is better?

2. To what extent the model and knowledge should be mutually adaptive?

3. How does various knowledge perform in few-shot settings?

Extensive experimental results and analysis demonstrate that: (1) Different forms of knowledge have their advantages in different situations of KILT. Specifically, the knowledge graph outperforms generation quality and exhibits stronger generalizability, while the knowledge text outperforms factual consistency in generations. Performance can be effectively improved by denoising the knowledge, for example, by selecting the succinct sequence or extracting a structured knowledge graph. (2) Performance could be universally improved further by advanced Dual-Encoders structure or by employing domain adaption pre-training. However, the impact of model size is highly related to the knowledge's own characteristics. Future work should choose the model size according to the situation and explore the broader impact of knowledge's characteristics. (3) The number of samples affects the model selection. When the samples are extremely small (100–200 in our studies), it is preferable for the input form and model architecture to resemble the pre-trained model as much as possible. However, the model architecture selection criteria tend to favour the task itself as the amount of data increases (500+), as there is sufficient data for the model to do task-level adaptive learning.

The contribution of this paper is that it is, to the best of our knowledge, the first work to thoroughly investigate different forms of knowledge, knowledge graph and knowledge text, and their respective advantages and disadvantages in knowledge-grounded dialogues. Furthermore, we comprehensively analyze implications from model selection, knowledge representation, and how to adapt the capacity, highlighting feasible solutions and shedding light on future research and development.

## 2  Literature Review

Knowledge-grounded dialogue is the task of generating an informative response $\mathcal{R}$ based on both dialogue history $\mathcal{C}$ and external knowledge $\mathcal{K}$. It is becoming an increasingly important topic in our community. The involved knowledge can be in different forms, such as documents for open-domain conversation (Dinan et al., 2019), movie database for movie recommendation conversation (Li et al., 2018), and knowledge graph for recommendation conversation (Liu et al., 2021b).

Recent research investigates several techniques for improving knowledge representation to fuse

knowledge in response generation. Ghazvinine-jad et al. (2018) encoded the dialogue history and knowledge documents separately to infuse the response with external world facts. Wang et al. (2022) added a particular knowledge graph representation in the response generation module.

Another line of work investigates pre-training methods for the knowledge-grounded dialogue to improve the system's performance on unseen topics and train in a low-resource setting. Zhao et al. (2020a) pre-trained the dialogue generation module and the knowledge encoder with ungrounded dialogues and the Wikipedia dump separately. Li et al. (2020) devised a variational network that can effectively estimate a generation model from a dialogue corpus and an independent knowledge corpus. Liu et al. (2021a) proposed a three-stage learning framework to separately pre-train the dialogue module and knowledge module.

Inspired by the accomplishments of pre-trained language models (PLM) for a range of natural language processing (NLP) tasks, researchers investigate learning knowledge through PLM' parameters. Zhao et al. (2020b) equipped a pre-trained language model with a knowledge selection module and presented an unsupervised jointly optimizing technique. Prabhumoye et al. (2021) focused on the attention mechanism in Transformers and proposed a strong baseline by a pre-trained model. Rashkin et al. (2021) increased the faithfulness in knowledge-grounded dialogue with controllable features of pre-trained models.

However, the above studies only considered a single form of knowledge, neglecting the impact of knowledge forms. Chen et al. (2020) proposed a pre-trained model for data-to-text tasks by unifying the knowledge format in the pre-training data and downstream tasks. Li et al. (2022) used unified essential knowledge triples instead of documents in pre-training, making their model more generalizable. Motivated by these methods, we conduct a thorough investigation of knowledge-grounded dialogue in various knowledge forms to explore the impact of knowledge. We focus on how to choose a knowledge form (sequence or graph) for NLG and investigate questions about which knowledge form is better and how it affects performance. Our study will facilitate the subsequent data collection and model construction and even influence the direction of knowledge-intensive language tasks.

## 3 Experimental Settings

In order to investigate the effect of various forms of knowledge, we comprehensively analyze the performance of different knowledge. Further, we also consider the factors of models to identify the interrelation between models and knowledge. This section introduces the detailed experimental settings for a clearer illustration[1].

### 3.1 Datasets

Our experiments were carried out on three public datasets, WoW (Dinan et al., 2019), Faith-Dial (Dziri et al., 2022a) and OpenDialKG (Moon et al., 2019). Table 1 shows the data statistics.

**WoW** The dataset consists of human-human conversations collected over Amazon Mechanical Turk and is grounded in Wikipedia passage (Dinan et al., 2019). These conversations are grounded in a diverse range of topics which are further split into seen and unseen topics during training and validation. The dialogues between two humans, referred to as "Wizard" and "Apprentice", engage in a conversation about a given knowledge. The dialogue is designed to be coherent and informative, with the Wizard providing information about the topic and the Apprentice asking questions and seeking clarification. The original knowledge in WoW is passages retrieved from Wikipedia, and we extracted triples by Open Information Extraction (OpenIE) annotator[2] as a knowledge graph. Since we consider constructing a knowledge-grounded agent, only the utterances from Wizard are used.

**FaithDial** Dziri et al. (2022b) analyzed the hallucination phenomenon and found that more than 60% of the response are hallucinated in the three datasets (including WoW). To mitigate this behavior, Dziri et al. (2022a) create FaithDial for hallucination-free dialogues by editing hallucinated responses in the WoW. As the FaithDial is an edited version of WoW, the settings of FaithDial are the same as WoW. The original knowledge in FaithDial is one sentence from Wikipedia, which is shorter and more accurate, and we also construct a knowledge graph by OpenIE as in WoW.

**OpenDialKG** OpenDialKG dataset contains conversations between two crowdsourcing agents engaging in a dialogue about a given topic. Open-

---

[1]we will release our code and data in the future.
[2]https://nlp.stanford.edu/software/openie.html

| Dataset | Form | Avg. Turns | Train Uttrs | Dev Uttrs | Test Uttrs | Avg. Tokens | Avg. Nodes |
|---|---|---|---|---|---|---|---|
| WoW | Sequence | 9.0 | 30.5K | 3.5K | 3.5K | 141.63 | 48.55 |
| FaithDial | Sequence | 9.0 | 11.2K | 2.2K | 2.3K | 26.36 | 17.92 |
| OpenDialKG | Graph | 5.8 | 14.8K | 3.2K | 3.2K | 12.52 | 3.56 |

Table 1: Data statistics of Wow, FaithDial and OpenDialKG. *Form* denotes the original forms of knowledge in datasets. *Avg. Turns* indicates the average number of turns involved in the dialogue. *Avg. Tokens* and *Avg. Nodes* indicate the size of different types of knowledge. The *Uttrs* denotes the number of total used utterances in datasets.

DialKG annotates a knowledge graph path label for each dialogue and a triple label for each dialogue turn. The response is grounded on the labeled triples during data collection. The original knowledge in OpenDialKG is one or some triples in the knowledge graph, and the sequence reasoning path is used as the source text. For example, the original triple is *<"Ryan Reynolds", " starred_actors", "Turbo">* and sequence reasoning path is *"Ryan Reynolds starred in Turbo"*.

## 3.2 Architectures

We utilize three typical architectures in NLG for a thorough comparison.

**Decoder-Only** Decoder-Only architecture is a typical language model characterized by a single autoregressive decoder. The decoder relies on previous tokens within a sequence as input to predict the subsequent token, following the conditional probability distribution $(p(x_t|x_{<t}))$. It generates text from left to right and is, therefore, able to generate human-like text by predicting the most likely next token at each time step. we employ GPT-2 as the Decoder-Only model. We concatenate the knowledge, the dialogue history, and the response as the input, i.e., $p(\mathcal{R}_t|[\mathcal{K};\mathcal{C};\mathcal{R}_{<t}])$, where $[;]$ denotes the concatenation of strings.

**Encoder-Decoder** Encoder-Decoder is another representative paradigm for NLG (Lewis et al., 2019; Raffel et al., 2020). Unlike Decoder-Only models, the Encoder-Decoder architecture models sequence-to-sequence generation by combining a bidirectional encoder and an autoregressive decoder. In our experiments, to individually model the different semantic spaces of knowledge and dialogue, we only pass the knowledge $\mathcal{K}$ to the encoder. The dialogue history $\mathcal{C}$ is considered as the prompt of the decoder, i.e., $p(\mathcal{R}_t|\mathcal{K}, [\mathcal{C};\mathcal{R}_{<t}])$, where $[;]$ denotes the concatenation of strings. We use BART (Lewis et al., 2019) as our backbone.

**Dual-Encoders** Dual-Encoders is a remarkable architecture for knowledge-grounded dialogue generation which encodes knowledge and dialogue history by two encoders and fuse them with attention mechanism in decoder (Prabhumoye et al., 2021; Liu et al., 2021a; Yang et al., 2022). Inspired by these works, we adopt Dual-Encoders architecture in our experiment. The knowledge encoder encodes the knowledge $\mathcal{K}$, and the dialogue encoder encodes dialogue history $\mathcal{C}$. Each layer of the decoder contains a self-attention block, a context cross-attention attending to dialogue context $\mathcal{C}$ and a knowledge cross-attention attending to knowledge $\mathcal{K}$, i.e., $p(\mathcal{R}_t|\mathcal{K}, \mathcal{C}, \mathcal{R}_{<t})$.

There maybe many other variants to improve these basic architectures. However, the most simplest configuration demonstrate the most fundamental abilities of the model. This kind of analysis provides the most important reference for other researchers. Adding too many components may distract from the main focus of the paper and make it difficult for readers to understand the key contributions and findings. Besides, as the parameters of different architecture is much different, we only focus on the impact from knowledge form and avoid influences from the number of model parameters as much as possible.

## 3.3 Serialize Knowledge Graph

The above architectures are based on the Transformers (Vaswani et al., 2017), which has become the dominant model in the field of NLP. Such models can effectively handle discrete sequences of tokens rather than graphs. Hence, we serialize the knowledge graph as input to the model. Specially, we add three special tokens, "*[triple]*", "*[entity]*", and "*[relation]*", to identify the triple, entity, and relation, respectively. For example, "*<guy ritchie, written by, snatch>, <snatch, starred actors, ewen bremner>*" is a knowledge graph with two triples. It will be serialized as "*[triple] [entity] guy ritchie [relation] written by [entity] snatch [triple] [entity] snatch [relation] starred actors [entity] ewen*

| Architecture | Knowledge | BLEU-2 | BLEU-4 | ROUGE-2 | ROUGE-L | DIST-1 | DIST-2 |
|---|---|---|---|---|---|---|---|
| | | | OpenDialKG | | | | |
| Decoder-Only | Sequence | 26.60 | 15.09 | 20.09 | 33.13 | 8.92 | 33.92 |
| | Graph | 27.13 | 15.62 | 21.07 | 34.18 | 9.28 | 35.01 |
| Encoder-Decoder | Sequence | 25.08 | 14.37 | 19.34 | 31.30 | 10.00 | 37.42 |
| | Graph | 25.57 | 14.28 | 19.14 | 31.41 | 9.68 | 35.85 |
| Dual-Encoders | Sequence | 27.35 | 15.12 | 20.55 | 34.12 | 9.65 | 37.66 |
| | Graph | 27.70 | 15.57 | 20.45 | 34.34 | 10.46 | 35.55 |
| | | | FaithDial Seen | | | | |
| Decoder-Only | Sequence | 25.07 | 12.61 | 16.50 | 32.18 | 10.43 | 40.26 |
| | Graph | 18.98 | 8.15 | 11.60 | 27.40 | 10.63 | 42.23 |
| Encoder-Decoder | Sequence | 24.47 | 12.21 | 18.25 | 34.41 | 9.64 | 40.07 |
| | Graph | 20.60 | 9.23 | 13.71 | 28.97 | 10.40 | 43.89 |
| Dual-Encoders | Sequence | 27.59 | 14.51 | 18.29 | 34.43 | 9.62 | 33.29 |
| | Graph | 21.16 | 9.68 | 14.51 | 30.57 | 10.59 | 38.42 |
| | | | FaithDial Unseen | | | | |
| Decoder-Only | Sequence | 23.60 | 10.69 | 15.21 | 31.41 | 15.88 | 54.25 |
| | Graph | 18.96 | 7.91 | 12.01 | 27.83 | 14.62 | 51.46 |
| Encoder-Decoder | Sequence | 24.02 | 10.88 | 17.35 | 33.57 | 14.52 | 54.36 |
| | Graph | 21.42 | 9.06 | 13.58 | 28.62 | 14.05 | 54.40 |
| Dual-Encoders | Sequence | 24.17 | 10.28 | 15.31 | 31.89 | 16.60 | 59.41 |
| | Graph | 20.38 | 8.72 | 14.55 | 30.55 | 15.92 | 52.16 |
| | | | WoW Seen | | | | |
| Decoder-Only | Sequence | 14.37 | 7.12 | 8.21 | 20.89 | 13.13 | 54.29 |
| | Graph | 15.68 | 7.16 | 9.48 | 22.57 | 12.34 | 53.19 |
| Encoder-Decoder | Sequence | 13.73 | 6.89 | 8.76 | 20.35 | 11.53 | 50.55 |
| | Graph | 17.31 | 9.03 | 11.00 | 22.27 | 11.62 | 51.87 |
| Dual-Encoders | Sequence | 16.00 | 8.20 | 9.78 | 22.88 | 14.21 | 58.70 |
| | Graph | 20.31 | 11.02 | 14.75 | 28.83 | 12.91 | 53.78 |
| | | | WoW Unseen | | | | |
| Decoder-Only | Sequence | 13.69 | 6.92 | 7.86 | 20.72 | 8.93 | 41.44 |
| | Graph | 13.85 | 5.48 | 7.67 | 21.39 | 9.13 | 44.44 |
| Encoder-Decoder | Sequence | 13.00 | 6.67 | 8.37 | 19.90 | 7.61 | 36.27 |
| | Graph | 15.00 | 7.14 | 9.09 | 20.46 | 8.68 | 43.53 |
| Dual-Encoders | Sequence | 15.39 | 7.81 | 9.19 | 22.17 | 9.22 | 39.16 |
| | Graph | 18.60 | 9.47 | 12.63 | 26.80 | 9.18 | 42.41 |

Table 2: The response quality performance of large pre-trained models.

*bremner*". Considering the knowledge graph is disordered, we sort the triples according to the token overlap between the triples and the dialogue history, which keeps the potential knowledge triples from being truncated. According to our experiments, as long as the valuable triples can be within the maximum length, the effect of the order is not significant. In this way, we bridge the information granularity gap between graph nodes and text tokens, i.e., a node is composed of many tokens. Furthermore, we can leverage a pre-trained model for the graph by considering the serialized knowledge graph as a natural language.

### 3.4 Metrics

Knowledge-grounded dialogue generation involves two-fold evaluations: first, it needs to generate coherent and diverse responses as a dialogue generation task. Second, as a knowledge-intensive task, the generation needs to satisfy factual consistency with external knowledge. To evaluate the quality of the response, we adopt **BLEU** (Papineni et al., 2002) and **ROUGE** (Lin, 2004) to evaluate the generated response against the reference. Higher scores indicate that the generated results are closer

to the reference. We also evaluate the diversity of generated response by n-grams distinct(Li et al., 2016). Following previous works in evaluating faithfulness (Dziri et al., 2022a), we take the **NLI**, $Q^2$**F1** and $Q^2$**NLI** (Honovich et al., 2021) as metrics. The **NLI** evaluates the text entailment, and the $Q^2$evaluates the factual consistency by question generation and question answering.

## 4 Experimental Analyses

This section presents our detailed investigation of how the knowledge forms affect the knowledge-grounded dialogue, focusing on the better forms of knowledge, the mutual adaption of model and knowledge, and the performance under few-shot settings. In this section, large pre-trained models (GPT2-Large and BART-Large) are leveraged for initialization unless otherwise highlighted. We use sequence- or graph-based models to refer to models that use knowledge text or knowledge graph as knowledge, respectively.

To facilitate the following analyses, we highlight the knowledge's characteristics of different datasets. OpenDialKG provides manually annotated knowledge graphs and knowledge text, which are short and accurate knowledge. FaithDial and WoW both provide knowledge text and the corresponding knowledge graphs are extracted by automated tools. For knowledge text, FaithDial's knowledge is sentence-level knowledge annotated by humans, while the knowledge in WoW ranges over long passages retrieved from Wikipedia. In comparison, the extracted knowledge graph in both FaithDial and WoW is not as good quality as manually annotated OpenDialKG. Therefore, the results of graph-based models on OpenDialKG can be regarded as the upper-bound of knowledge graph grounded dialogue and the results of sequence-based models on FaithDial and OpenDialKG can be regarded as the upper-bound of knowledge text grounded dialogue. However, as retrieving the precise knowledge sentence or triples is difficult in real-world practice, the results on WoW is much closer to reality.

### 4.1 Q1: Graph v.s. Sequence, which form of knowledge form is better?

Our analyses examine the validity of the knowledge text and graph in knowledge-grounded dialogue and attempt to determine the better form.

| Architecture | Knowledge | NLI | $Q^2$ F1 | $Q^2$ NLI |
|---|---|---|---|---|
| OpenDialKG | | | | |
| Decoder-Only | Sequence | 50.69 | 42.20 | 41.56 |
| | Graph | 51.85 | 44.13 | 43.56 |
| Encoder-Decoder | Sequence | 53.41 | 43.89 | 41.91 |
| | Graph | 51.16 | 40.17 | 38.82 |
| Dual-Encoders | Sequence | 52.39 | 46.28 | 45.89 |
| | Graph | 55.42 | 44.14 | 42.05 |
| Faithdial Seen | | | | |
| Decoder-Only | Sequence | 62.27 | 59.15 | 54.31 |
| | Graph | 54.82 | 41.01 | 36.42 |
| Encoder-Decoder | Sequence | 64.28 | 67.03 | 61.44 |
| | Graph | 53.43 | 44.42 | 39.51 |
| Dual-Encoders | Sequence | 63.49 | 70.88 | 66.67 |
| | Graph | 56.70 | 49.34 | 46.04 |
| Faithdial Unseen | | | | |
| Decoder-Only | Sequence | 64.85 | 61.31 | 56.35 |
| | Graph | 58.70 | 45.89 | 41.33 |
| Encoder-Decoder | Sequence | 67.55 | 65.21 | 59.67 |
| | Graph | 60.85 | 49.37 | 44.31 |
| Dual-Encoders | Sequence | 69.34 | 68.14 | 61.39 |
| | Graph | 61.17 | 51.07 | 46.55 |
| WoW Seen | | | | |
| Decoder-Only | Sequence | 54.07 | 55.08 | 49.37 |
| | Graph | 53.42 | 41.24 | 36.35 |
| Encoder-Decoder | Sequence | 53.49 | 59.67 | 53.22 |
| | Graph | 52.62 | 42.60 | 37.08 |
| Dual-Encoders | Sequence | 54.16 | 66.65 | 58.94 |
| | Graph | 53.68 | 45.34 | 40.32 |
| WoW Unseen | | | | |
| Decoder-Only | Sequence | 53.82 | 55.55 | 50.46 |
| | Graph | 52.89 | 36.37 | 32.33 |
| Encoder-Decoder | Sequence | 53.22 | 60.87 | 55.17 |
| | Graph | 53.13 | 42.19 | 38.22 |
| Dual-Encoders | Sequence | 54.90 | 68.20 | 62.07 |
| | Graph | 53.99 | 43.73 | 39.31 |

Table 3: The faithful consistency performance of large pre-trained models.

**Response Quality** Table 2 illustrates the results of response quality. The results on OpenDialKG are numerically significantly higher than the other two data sets which may indicate that whether the form of knowledge is sequence or graph, more precise information will be more advantageous to the results. On the other hand, the performance of graph-based models is slightly higher than the sequence-based, which shows the upper-bound of graph-based models is better. While on FaithDial, the sequence-based models show a clear advantage as the knowledge graph is much noisier or sparse. The results of WoW present the opposite conclusion from FaithDial, where the graph-based models' performance is significantly better than that of the sequence-based model. It may indicate that when the knowledge text is complex and noisy, the extracted graph can act as an information filter to reduce the noise. This result provides us with

two paths to improve generation quality, one is to simplify the text, such as annotating fine-grained sentences, and the other is to extract the knowledge graph to reduce the redundancy.

Our observations indicate that in the FaithDial dataset, knowledge graphs improve response diversity in the "seen" setting, while knowledge text enhances diversity in the "unseen" setting. Conversely, in the WoW dataset, knowledge text improves diversity in the "seen" setting, whereas knowledge graph improves it in the "unseen" setting. These findings suggest that lower-quality knowledge can lead to greater diversity when the domain/topic is familiar during training, while higher-quality knowledge may be more beneficial when the domain/topic is unfamiliar. Furthermore, we speculate that the variation in diversity across datasets and knowledge formats may be attributed to differences in modeling difficulty and generalization capabilities. For instance, Decoder-Only architecture can readily model fine-grained sentences and generalize to unseen examples in FaithDial but may struggle with complex knowledge passages in WoW.

**Factual Consistency** The results of factual consistency are shown in Table 3. Unlike the response quality, the knowledge text shows an absolute advantage in factual consistency. It may be a bias that the metrics are calculated based on the text, which is closer to the sequence than the graph. However, we lack unified metrics to evaluate the factual consistency of the knowledge graph and sequence. Overall, the Dual-Encoders architecture outperforms both metrics and is a superior model for knowledge-intensive tasks.

**Generalizability** Dziri et al. (2022a) examines the usefulness of FaithDial in an out-of-domain setting by testing the performance on other datasets of models trained on FaithDial. Following the setting, we examine the generalizability of models with different types of knowledge by transferring the results of response generation from FaithDial to WoW. As shown in Figure 1, the knowledge graph results in response quality metrics are much higher than the knowledge text in all three architectures. The results on the unseen test set of FaithDial and WoW also support the observation. It demonstrates that graph-based models have stronger generalizability than sequence-based models, and when our data is limited, it is helpful to construct knowledge

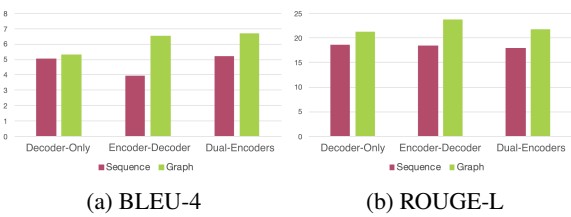

| (a) BLEU-4 | (b) ROUGE-L |

Figure 1: Generalization performance in terms of training on FaithDial and testing on WoW.

of graph-based data.

**Human Evaluation** In addition to reference-based metrics, comprehensive human evaluation is crucial for dialogue-related generation tasks. To this end, we randomly selected 100 samples with two responses (from sequence- or graph-based models) from each dataset for each architecture, and instructed annotators to choose the preferred response based on dialogue history and given knowledge. The results, presented in Table 4, indicate that sequence-based models outperform graph-based models, with a significantly higher win rate on FaithDial and WoW. These findings suggest that sequence-based models, particularly those using Dual-Encoders architecture, have a distinct advantage in interacting with humans. Furthermore, the results reveal a discrepancy between the reference-based metrics such as BLEU and ROUGE and human preferences.

| Dataset | Architecture | Win Rate | |
| --- | --- | --- | --- |
| | | Sequence | Graph |
| | Decoder-Only | 51% | 49% |
| OpenDialKG | Encoder-Decoder | 50% | 50% |
| | Dual-Encoders | 69% | 31% |
| | Decoder-Only | 66% | 34% |
| FaithDial Seen | Encoder-Decoder | 75% | 25% |
| | Dual-Encoders | 78% | 22% |
| | Decoder-Only | 65% | 35% |
| FaithDial Unseen | Encoder-Decoder | 84% | 16% |
| | Dual-Encoders | 76% | 24% |
| | Decoder-Only | 71% | 29% |
| WoW Seen | Encoder-Decoder | 75% | 25% |
| | Dual-Encoders | 75% | 25% |
| | Decoder-Only | 60% | 40% |
| WoW Unseen | Encoder-Decoder | 74% | 26% |
| | Dual-Encoders | 71% | 29% |

Table 4: Human comparison evaluation.

Based on the above analyses, for knowledge-grounded dialogue, the knowledge graph has better response quality and generalizability; however, the knowledge text is better in the factual consistency of generating responses. Moreover, manual

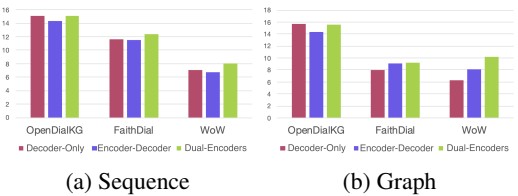

(a) Sequence    (b) Graph

Figure 2: BLEU-4 score of different architectures.

subgraphs selection in OpenDialKG and manually grounded sentence annotation in FaithDial inspire us that denoising the source knowledge can improve the generation. Specifically, we can finely retrieve an accurate subgraph when the source is the knowledge graph. While the source is the knowledge text, we can extract knowledge graphs to reduce redundancy like WoW or rank fine-grained semantic units to select precise knowledge.

## 4.2 Q2: To what extent the model and knowledge should be mutually adaptive?

Following the selection of the knowledge form, problems arise like as what kind of architecture is suitable for encoding the particular knowledge, whether a larger model size is preferable, whether a pre-trained model is required, etc. To facilitate the subsequent research, we comprehensively compare detailed factors of the model and investigate to what extent the model and knowledge should be mutually adaptive.

**Model Architecture**   Figure 2 compares the BLEU-4 score of different architectures. Besides OpenDialKG, the Decoder-Only architecture outperforms the Encoder-Decoder architecture for knowledge text, while the opposite is observed for the knowledge graph. On the contrary, the Dual-Encoders outperform others in most cases whatever the knowledge form is. It indicates that the Dual-Encoders have a stably excellent performance over varying forms of knowledge. Therefore, we conclude that the Dual-Encoders is a wise choice to eliminate the impact of different knowledge.

**Model Size**   Figure 3 compares the BLEU-4 score of various model sizes of Dual-Encoders for the two forms of knowledge. Hoffmann et al. (2022) stated that the model size of large language models should be matched to the size of the training data. In addition, our results further indicate that the model size should be mutually adaptive to the knowledge's characteristics. For instance, the

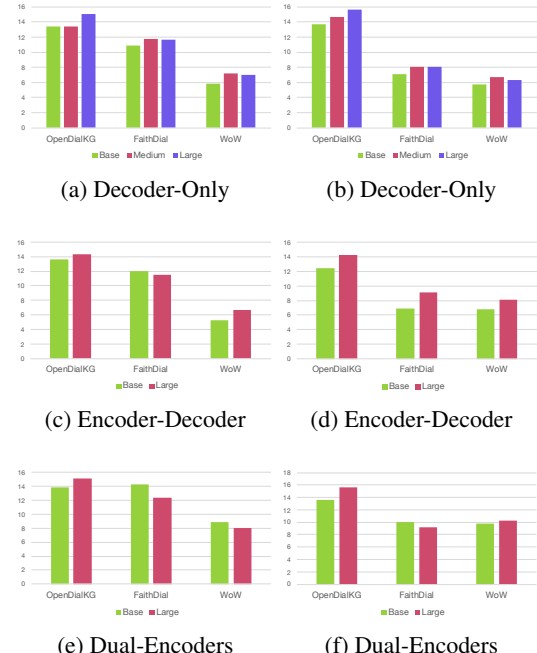

(a) Decoder-Only    (b) Decoder-Only

(c) Encoder-Decoder    (d) Encoder-Decoder

(e) Dual-Encoders    (f) Dual-Encoders

Figure 3: BLEU-4 score of various model size. The figures on the left side are based on result grounded in knowledge text, while the figures on the right side are based on result grounded in a graph.

knowledge in OpenDialKG is of high quality; even though it has limited data, the larger model performs better. We infer that this is because some knowledge's characteristics, such as the standard knowledge graph in OpenDialKG, are more conducive to exploring the potential of pre-trained models, like prompt learning. Furthermore, knowledge in WoW has higher noise for both sequence and graph forms, with varying effects on model size. The above statistics indicate it is not that the larger the model size is, the better performance is. The optimal model size is related to training data size and is highly associated with knowledge's sophisticated characteristics.

**Pretrained or not**   We experimented with investigating the impact of pre-trained initialization, focusing on the effect of different knowledge forms and architectures. Figure 5 shows the effect of pre-trained across different architectures and knowledge forms, with red indicating the results without pre-trained and blue indicating the improved results achieved with pre-trained. The positive impact of pre-trained initialization is universal. However, pre-trained can lead to a more remarkable performance improvement when the knowledge is represented in sequence form, compared to the serialization

form of the knowledge graph. The reason may be that the knowledge graph, a form of structured data, lacks the natural language pattern after serialization, making it less intuitive for pre-trained models designed to understand and generate language fluently. By comparing the results without pre-trained, we can see that the architectures exhibit different performances under different knowledge forms. For instance, the Dual-Encoders architecture yields the best performance in the sequence form, whereas the Decoder-Only architecture attains the optimal results in the graph form.

In terms of overall results, the Dual-Encoders architecture with pre-trained initialization positively impact the results. It suggests that performance could be improved further by advanced Dual-Encoders structure or by employing domain adaption pre-training, and the improvement is slightly related to knowledge's characteristics. Therefore, we suggest further research to choose Dual-Encoders with pre-trained initialization as a baseline and design experiments of various model sizes for different knowledge to indicate the contribution.

### 4.3 Q3: How does various knowledge perform in few-shot settings?

Since the knowledge-auxiliary is an important direction of few-shot learning (Chen et al., 2021), we would like to explore the performance of various knowledge-based models under few-shot settings. Unlike related work dedicated to few-shot learning (Li et al., 2022, 2020), we do not consider particular methods such as domain pre-training with pseudo-data to optimize our models, but only the models mentioned in the above paper.

Figure 4 shows the results from zero-shot to 1000-shot. Because the pre-trained model is trained on natural language, the sequence-based model outperforms the graph-based model. However, it may be due to the gap between serialized graphs and natural language for the pre-trained language model. The graph-based models would be better if the model employed a graph-based pre-training stage. Similarly, the Encoder-Decoder is closest to the pre-training phase, while the Dual-Encoders have enormous variation. Thus, the Encoder-Decoder performs best, while the Dual-Encoders require more training data to fine-tune. However, as the data increases, the Dual-Encoders' performance improves significantly, surpassing other models, such as 500-shot on OpenDialKG. Therefore, when we have

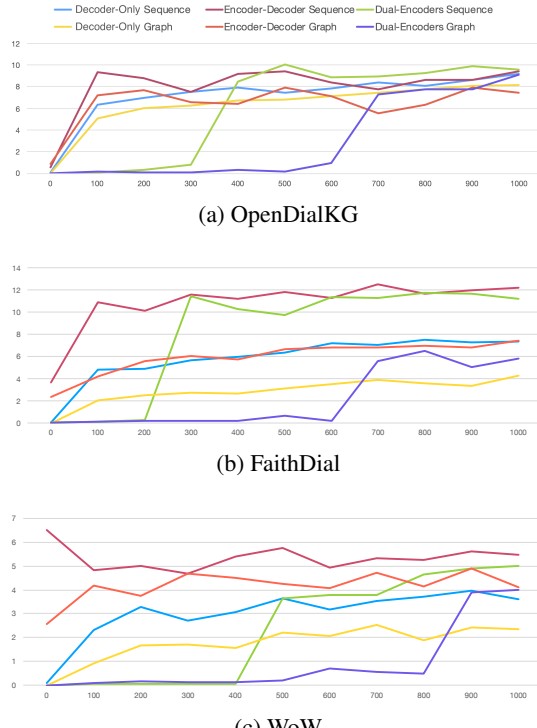

(a) OpenDialKG

(b) FaithDial

(c) WoW

Figure 4: BLEU-4 score in few-shot settings

only a few samples (100-200), the input form and model architecture should resemble the pre-trained model, such as Encoder-Decoder. Then, when the data is larger (500+) or better (e.g., detailed manual annotations), we can train a task-adapted model, such as Dual-Encoders.

## 5 Conclusion

In the paper, we analyze the strength and limitations of two forms of knowledge for knowledge-grounded dialogue. We conduct a comprehensive study to determine which form of knowledge is superior and to what extent the model and knowledge should be mutually adaptable. Additionally, we investigate the performance of various knowledge in few-shot settings. Our experimental results indicate that different knowledge forms have their advantages, but task-oriented instructions, such as proper knowledge and model selection, are essential. We also summarize determinant conclusions about the influencing factors for subsequent research. In future work, we will analyze the influencing factors more quantitatively and explore general conclusions for other knowledge-intensive tasks.

## Limitations

This work may contribute to the development of knowledge-intensive language tasks. Nevertheless, there are two main limitations of the works. First, limited by the experimental equipment, we can not access the larger pre-trained language models, such as T5-XXL, for the scaling experiments. As analyzed in Section 4.2, the model's size and knowledge are correlated. Therefore, all the conclusions in this paper could have changed if the larger models had been used. However, it is essential to realize that the training cost of large models is vast, so it may be necessary to compare effectiveness and efficiency. Secondly, We only focus on the knowledge-grounded dialogue generation as a representative task for knowledge-intensive language tasks. Besides, other tasks, such as fact-checking and question-answering, should be considered. The larger models will be considered in future work, and more extensive research will be introduced in other knowledge-intensive language tasks.

## Acknowledgements

We extend our sincere gratitude to the anonymous reviewers for their helpful feedback, the diligent efforts of the conference committees, and the collaboration of the members of the Beijing Engineering Research Center of High-Volume Language Information Processing and Cloud Computing Applications for their valuable insights. This work has received partial funding from the Joint Funds of the National Natural Science Foundation of China (Grant No. U21B2009) and the Major Research Plan of the National Natural Science Foundation of China (Grant No.92370110).

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

## A Implementation Details

Models in the paper were implemented by PyTorch and Transformers (Wolf et al., 2019). The Dual-Encoders and Encoder-Decoder architecture are based on BART (Lewis et al., 2019), and the Decoder-Only architecture is based on GPT-2 (Radford et al., 2019).

We used the transformer toolkit (Wolf et al., 2019) to implement the models. We initialized the external parameters for the Dual-Encoders following DoHA (Prabhumoye et al., 2021). Specifically, We initialized the two encoders with the

| Architecture | Sequenceize | Params |
|---|---|---|
| Decoder-Only | base | 237.35M |
| | Medium | 676.77M |
| | Large | 1476.35M |
| Encoder-Decoder | Base | 265.92M |
| | Large | 774.94M |
| Dual-Encoders | Base | 375.60M |
| | Large | 1161.39M |

Table 5: The number of parameters of models in different size.

same weights for the Dual-Encoders model, and the two cross-attention blocks were initialized with the same weights in the decoder layers. Hence, the layer size of the cross-attention is the same as in BART. We trained all models for 10k steps, using a batch size of 32 and the Adam optimizer (Kingma and Ba, 2014) with a learning rate of 5e-5. We warmed up the learning rate for 0.5k steps followed by a linear decay. The max length of dialogue history and response is 128 and 64 tokens. The max length of the knowledge text in OpenDialKG, FaithDial, and WoW are 64, 64, and 256. The knowledge graph's max length is 64, 512, and 512, respectively. The models were evaluated on the validation set per 0.5k steps, and the best-performing model was saved for testing. We early stopped the training with patience of 5. Training for all models was done on an Nvidia GeForce RTX 3090GPU 24GB, and for inference, we used nucleus sampling with p=0.6. Table 5 shows the statistics of parameters of models in different sizes.

For few-shot settings, inspired by the experiments on model size, we only explore the performance of base-size models (BART-base and GPT2-base) with pre-trained initialization. We also experiment with the large-size model, but the results show that the large-size model could not fit the few-shot data. To prevent overfitting at the few-shot setting, we reduce the learning rate to 1e-5, set the warmup steps to 100, and set the total training steps to 5K. We also early stop the training with a patience of 3.

### A.1 Impact of Pretrained

## B Encode Graph by GAT

In addition to serialization, we also considered leveraging GAT (Veličković et al., 2018) to encode the knowledge graph, and the results are shown in Table 6. Specifically, we pool the token representa-

| Architecture | dataset | Graph Encoder | Pre-trained or not | BLEU-2 | BLEU-4 | ROUGE-2 | ROUGE-L | DISTINCT-1 | DISTINCT-2 |
|---|---|---|---|---|---|---|---|---|---|
| Encoder-Decoder | OpenDialKG | GAT-Encoder | No | 12.41 | 5.03 | 8.43 | 19.93 | 4.40 | 21.98 |
| | | Serialization | No | 13.33 | 5.81 | 7.61 | 16.80 | 6.60 | 28.55 |
| | | | Yes | 25.57 | 14.28 | 19.14 | 31.41 | 9.68 | 35.85 |
| | FaithDial Seen | GAT-Encoder | No | 6.90 | 1.92 | 3.30 | 14.10 | 7.38 | 38.17 |
| | | Serialization | No | 6.24 | 1.57 | 3.05 | 13.64 | 8.06 | 39.60 |
| | | | Yes | 20.60 | 9.23 | 13.71 | 28.97 | 10.40 | 43.89 |
| | FaithDial Unseen | GAT-Encoder | No | 9.03 | 2.57 | 4.85 | 16.70 | 7.97 | 41.48 |
| | | Serialization | No | 8.39 | 2.67 | 4.47 | 15.52 | 8.36 | 41.48 |
| | | | Yes | 21.42 | 9.06 | 13.58 | 28.62 | 14.05 | 54.40 |
| | Wow Seen | GAT-Encoder | No | 6.04 | 1.17 | 2.71 | 13.89 | 5.98 | 35.60 |
| | | Serialization | No | 5.66 | 1.10 | 2.29 | 12.24 | 5.91 | 38.59 |
| | | | Yes | 17.31 | 9.03 | 11.00 | 22.27 | 11. 62 | 51.87 |
| | WoW Unseen | GAT-Encoder | No | 4.52 | 0.63 | 1.75 | 11.99 | 5.68 | 33.65 |
| | | Serialization | No | 5.08 | 0.76 | 1. 65 | 12.88 | 8.20 | 41.52 |
| | | | Yes | 15.39 | 7.81 | 9.19 | 22.17 | 9.22 | 39.16 |
| Dual-Encoder | OpenDialKG | GAT-Encoder | No | 15.04 | 5.89 | 9.10 | 22.61 | 6.21 | 27.02 |
| | | Serialization | No | 13.72 | 5.44 | 8.73 | 22.00 | 6.55 | 28.26 |
| | | | Yes | 27.70 | 15.57 | 20.45 | 34.34 | 10.46 | 35.55 |
| | FaithDial Seen | GAT-Encoder | No | 8.70 | 2.21 | 3.80 | 17.09 | 10.80 | 46.41 |
| | | Serialization | No | 8.72 | 2.49 | 3.88 | 16.85 | 10.08 | 42.90 |
| | | | Yes | 21.16 | 9.68 | 14.51 | 30.57 | 10.59 | 38.42 |
| | FaithDial Unseen | GAT-Encoder | No | 11.74 | 3.87 | 6.24 | 20.28 | 11.89 | 50.50 |
| | | Serialization | No | 10.37 | 3.41 | 4.95 | 1 8.68 | 10.60 | 45.09 |
| | | | Yes | 20.38 | 8.72 | 14.55 | 30.55 | 15.92 | 52.16 |
| | Wow Seen | GAT-Encoder | No | 8.74 | 2.35 | 3.80 | 16.64 | 8.73 | 44.39 |
| | | Serialization | No | 8.85 | 2.12 | 3.93 | 16.97 | 8.60 | 44,.96 |
| | | | Yes | 20.31 | 11.02 | 14.75 | 28.83 | 12.91 | 53.78 |
| | WoW Unseen | GAT-Encoder | No | 5.87 | 1.01 | 2.09 | 13.87 | 8.47 | 42.51 |
| | | Serialization | No | 6.20 | 1.06 | 2.17 | 14.13 | 8.86 | 42.81 |
| | | | Yes | 18.60 | 9.47 | 12.63 | 26.80 | 9.18 | 42.41 |

Table 6: The comparison results of graph-based models with GAT encoder and serialization.

tion as a representation of the nodes and leverage GAT instead of the encoder in the architecture. As we do not apply pre-trained initialization on GAT, we only compare with the no pre-trained models. The table shows that the graph encoder results are better than those leveraging the serialization approach. However, the graph encoder models also underperform the models with pre-trained initialization. Therefore, when pre-training with Graph-to-Sequence is unavailable, it is an excellent way to serialize the knowledge graph into a sequence.

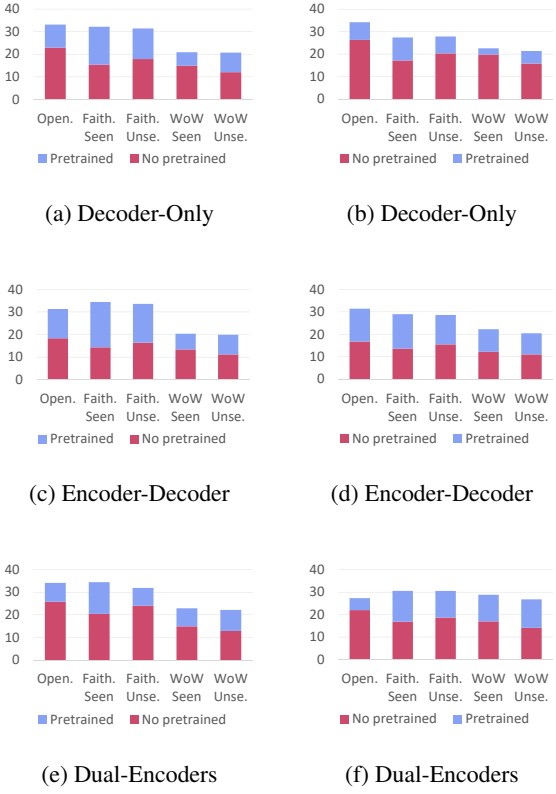

Figure 5: The effect of pretrained. The figures on the left side are based on result grounded in knowledge text, while the figures on the right side are based on result grounded in a graph.