# OpenReview forum: "Graph vs. Sequence: An Empirical Study on Knowledge Forms for Knowledge-Grounded Dialogue"
_EMNLP/2023/Conference — EMNLP 2023 Main_

### Official Review · Reviewer_E5Fq · 2023-08-05

**Soundness:** 4

**Excitement:**

2: Mediocre: This paper makes marginal contributions (vs non-contemporaneous work), so I would rather not see it in the conference.

**Paper Topic And Main Contributions:**

The work studies how knowledge graphs (knowledge tuples) compare with knowledge sequences (unstructured textual contents). The author's posit that the two forms of knowledge are to be studied "on their own" and their impact on performance in models of choice. The authors perform experiments on four popular and relevant datasets. To compare the performance on two types of knowledge representations, the authors convert the knowledge graph into sequence of tokens and leverage decoder-only, encoder-decoder and dual encoder models for their experiments. The authors list model-architecture recommendations based on the experiments performed.


**Reasons To Accept:**

The experiments performed to conclude the model architecture trade-offs are impressive. (Appendix is more interesting than the paper!)


**Reasons To Reject:**

- The main drawback of the paper is the writing. It is very difficult to follow the message.

For instance,
1. Some of the terminologies of choice are confusing. Some examples:
"Autoregressive": Are the authors talking about decoder-only archictures? BART is autoregressive as well.
"Knowledge Sequence": is there are any difference with the text sequence (Which is simpler and easier to discern).

2. The authors show the promise of dual encoder models but the numbers and conclusions strongly recommend dual encoder models. A statistical significance test for instance could help make a stronger case. From the numbers and metrics chosen, it is hard to conclude the superiority of the model recommended. The performance metrics are not the only choice for model selection often times. Scalability and the ability to be adapted to different applications (for instance, prompting & Instruction tuning enable development of various other applications).

3. The papers are supposed to not rely on Appendix to convey the message. A lot of the results from the appendix conveys the findings better. The writing needs to be significantly improved to get a lot of the interesting results from the Appendix into the main paper.


**Reproducibility:**

3: Could reproduce the results with some difficulty. The settings of parameters are underspecified or subjectively determined; the training/evaluation data are not widely available.

**Reviewer Confidence:**

4: Quite sure. I tried to check the important points carefully. It's unlikely, though conceivable, that I missed something that should affect my ratings.

---

> ### Author Rebuttal · Authors · 2023-08-29
>
> Thanks for your review and Suggestions. We apologize if the current writing style has made our paper difficult to follow. We take your feedback seriously and will strive to enhance the clarity and flow of our writing in the revised version, ensuring our message is effectively conveyed.
>
> - We apologize if our choice of terminologies has caused confusion. Indeed, "decoder-only" is a more accurate term than "auto-regressive," and "text sequence" is simpler and easier to understand than "knowledge sequence." Your corrections are well-received and will be carefully incorporated in our revisions. Thank you for your insightful suggestions.
>
> - We appreciate your insightful suggestions. In response, we have conducted a p-value significance test which confirms that the dual-encoder architecture statistically outperforms other architectures (p-value < 0.05). We concur that performance metrics alone are insufficient for model selection. Our study also explores other aspects like generalizability across different architectures and knowledge forms (refer to Section 4.1).
> Regarding scalability and adaptability, we performed experiments on large language models under zero-shot instruction, using Alpaca (13B) as a Decoder-Only architecture and Flan-T5-XXL (11B) as an Encoder-Decoder architecture. We validated the coherence and faithfulness of these large-scale models via human evaluations using Likert scores.
>
> |  Dataset   | Model  |  Knowledge Form | Coherence | Faithfulness |
> |  :----:  | :----:  |  :----:  |  ----:  |  ----:  |
> | OpenDialKG | Alpaca | text | 4.42 | 3.66 |
> | OpenDialKG | Alpaca | graph | 4.50 | 3.61 |
> | OpenDialKG | Flan-T5-xxl | text | 3.91 | 3.33 |
> | OpenDialKG | Flan-T5-xxl | graph | 3.60 | 3.48 |
> | FaithDial Seen | Alpaca | text | 4.24 | 3.25 |
> | FaithDial Seen | Alpaca | graph | 4.14 | 3.37 |
> | FaithDial Seen | Flan-T5-xxl | text | 3.41 | 3.43 |
> | FaithDial Seen | Flan-T5-xxl | graph | 2.99 | 3.47 |
> | FaithDial Unseen | Alpaca | text | 4.48 | 3.74 |
> | FaithDial Unseen | Alpaca | graph | 4.24 | 3.47 |
> | FaithDial Unseen | Flan-T5-xxl | text | 3.53 | 3.55 |
> | FaithDial Unseen | Flan-T5-xxl | graph | 3.14 | 3.36 |
> | WoW Seen | Alpaca | text | 3.93 | 4.32 |
> | WoW Seen | Alpaca | graph | 3.86 | 3.69 |
> | WoW Seen | Flan-T5-xxl | text | 3.61 | 3.75 |
> | WoW Seen | Flan-T5-xxl | graph | 3.28 | 3.20 |
> | WoW Unseen | Alpaca | text | 4.32 | 4.34 |
> | WoW Unseen | Alpaca | graph | 3.86 | 3.37 |
> | WoW Unseen | Flan-T5-xxl | text | 4.05 | 3.53 |
> | WoW Unseen | Flan-T5-xxl | graph | 3.88 | 2.80 |
>
> As evident from the results, larger models demonstrate a preference for text due to the extensive text data used during pre-training and instruction tuning. However, the conclusions stated in our paper, such as the impact of knowledge quality and architecture, still hold true. These findings suggest valuable insights for building knowledge-grounded conversation applications and exploring the potential of other architectures (like Dual-Encoders) when scaled up and adapted for tuning. Nonetheless, our analysis is mainly based on small-scale models.
>
> - Thank you for your constructive feedback. We acknowledge that a significant portion of our results is presented in the appendix due to the large number of experiments we conducted. Given the paper's space constraints, it was challenging to include all these results and analyses in the main body. Some key findings from the appendix include:
>
>     (1)The robust performance of Dual-Encoders architecture across varying forms of knowledge.
>
>     (2)The strong generalizability of graph-based models as indicated by our transfer experiments.
>
>     (3)Insights on the interplay among model size, knowledge form, and knowledge size, which can guide practical application.
>
>     (4)The superiority of pre-training initialized sequence-based models over models with graph-encoder for knowledge graph. This suggests that when Graph-to-Sequence pre-training is unavailable, serializing the knowledge graph into sequence for sequence-based pre-trained models is a good alternative.
>
>     Taking into account your suggestion, we will strive to optimize the presentation in our revised version. We'll balance the constraints of space while ensuring the most valuable and interesting results are included in the main text.

---

### Official Review · Reviewer_eXkB · 2023-08-06

**Soundness:** 4

**Excitement:**

4: Strong: This paper deepens the understanding of some phenomenon or lowers the barriers to an existing research direction.

**Paper Topic And Main Contributions:**

The contribution of this paper is that it investigate different forms of knowledge, KG and KS, and their respective advantages and disadvantages in knowledge-grounded dialogues.
The paper aims to answer three main questions:
1. Graph v.s. Sequence, which form of knowledge is better?
2. To what extent the model and knowledge should be mutually adaptive?
3. How does various knowledge perform in few shot settings?


**Questions For The Authors:**

1-	My question and suggestion concern noise since we know this can impact model structure. What you should do to minimize noise impact on model size?

**Reasons To Accept:**

1-	The paper is weel structured and present clearly the focos and main scientific goals;
2-	Experiments use three different architectures that produce good insigths for further research


**Reasons To Reject:**

1-	The text as too much abbreviations. Some times is hard to follow;

**Reproducibility:**

4: Could mostly reproduce the results, but there may be some variation because of sample variance or minor variations in their interpretation of the protocol or method.

**Reviewer Confidence:**

4: Quite sure. I tried to check the important points carefully. It's unlikely, though conceivable, that I missed something that should affect my ratings.

---

> ### Author Rebuttal · Authors · 2023-08-29
>
> Thanks for your review and Suggestions.
>
> - We apologize if the extensive use of abbreviations in our text made it challenging to follow. In our revised version, we will endeavor to minimize the use of abbreviations and provide clearer explanations where necessary, thereby enhancing readability and comprehension..
>
> - As analyzed in Section 4.2, high noise knowledge can indeed confuse larger models, particularly those based on Dual-Encoder Architectures. That is an important and interesting future work involved from our analyses. We hypothesize that this issue arises because such models aren't sufficiently pre-trained and thus are more sensitive to noisy knowledge input. To mitigate this, we suggest a post-training stage for larger pre-trained models to better tailor them to the task at hand. Moreover, as larger models inherently have higher capability, they are more sensitive to fitting or overfitting to noise in data. Therefore, another possible solution is to 'denoise' the training data or modify it to closely resemble the format of pre-training inputs. This could involve retrieving fine-grained text and improving information extraction technology, which may prove beneficial for larger model training.

---

### Official Review · Reviewer_Ez2J · 2023-08-07

**Soundness:** 3

**Excitement:**

3: Ambivalent: It has merits (e.g., it reports state-of-the-art results, the idea is nice), but there are key weaknesses (e.g., it describes incremental work), and it can significantly benefit from another round of revision. However, I won't object to accepting it if my co-reviewers champion it.

**Missing References:**

Search suggests several recent surveys exist on the topic; these are not mentioned in the paper.

Yu, Wenhao, et al. “A Survey of Knowledge-Enhanced Text Generation.” ACM Computing Surveys, vol. 54, no. 11s, Jan. 2022, pp. 1–38. Crossref, https://doi.org/10.1145/3512467.

Mialon, Grégoire, Roberto Dessì, Maria Lomeli, Christoforos Nalmpantis, Ram Pasunuru, Roberta Raileanu, Baptiste Rozière et al. "Augmented language models: a survey." arXiv preprint arXiv:2302.07842 (2023).

Chowdhury, T., Ling, C., Zhang, X., Zhao, X., Bai, G., Pei, J., Chen, H. and Zhao, L., 2023. Knowledge-enhanced Neural Machine Reasoning: A Review. arXiv preprint arXiv:2302.02093.

Pan, S., Luo, L., Wang, Y., Chen, C., Wang, J. and Wu, X., 2023. Unifying Large Language Models and Knowledge Graphs: A Roadmap. arXiv preprint arXiv:2306.08302.

Groth, P., Simperl, E., van Erp, M. and Vrandečić, D., 2023. Knowledge Graphs and their Role in the Knowledge Engineering of the 21st Century (Dagstuhl Seminar 22372). In Dagstuhl Reports (Vol. 12, No. 9). Schloss Dagstuhl-Leibniz-Zentrum für Informatik.


**Paper Topic And Main Contributions:**

This paper analyzes the strength and limitations of graph and sequence based representations of knowledge for knowledge grounded dialogue. They use 3 datasets and conduct studies to test effect of different representations on response quality, factual consistency, generalizability and response scores in few shot settings. The paper tries to compare graphs and sequences head to head and answer which for may be better under various circumstances.



**Reasons To Accept:**

The paper suggests some good findings as follows:
- the knowledge graph outperforms generation quality and exhibits stronger generalizability, while the knowledge sequence outperforms factual consistency in generations.
- Performance can be effectively improved by denoising the knowledge, for example, by selecting the succinct sequence or extracting a structured knowledge graph.
- Performance could be universally improved further by advanced Dual-Encoders structure or by employing domain adaption pre-training; however, impact of model size and knowledge size is understudied
- (and more)
The findings should be reported in a tabular form for quick access to the results and findings.
- Exhaustive set of experiments are reported in the appendix to study the 3 major questions authors are trying to study in this work.

**Reasons To Reject:**

- This seems like a limited study that uses 3 datasets for studying knowledge augmented response generation. The goal of the study is to compare graphs and text sequences. How one represents the text and graphs structures for knowledge augmentation including the reasoning methods utilized with the graphs is understudied (to make general claims about which paradigm is better)
- Dialog evaluation metrics are very dated and do not correlate well with human judgment.
- Response quality results seem to be inconclusive and dataset dependent
- As the authors note, factual consistency results may be biased due to the text based metrics used in the study
- Various graph embedding approaches exist, how would these interplay with the response generation system and compare to the sequence based representations.
- Role of prompting that is very common presently in response generation is not studied at all in the paper.

**Reproducibility:**

4: Could mostly reproduce the results, but there may be some variation because of sample variance or minor variations in their interpretation of the protocol or method.

**Reviewer Confidence:**

4: Quite sure. I tried to check the important points carefully. It's unlikely, though conceivable, that I missed something that should affect my ratings.

---

> ### Author Rebuttal · Authors · 2023-08-29
>
> Thanks for your review and suggestions.
>
> - In addressing the critique of limited dataset use, it is important to underscore the representativeness and real-world applicability of the three datasets adopted in our study. As detailed in Section 3.1, OpenDialKG and FaithDial incorporate human-annotated knowledge graph or text, respectively. These datasets embody pure, high-quality knowledge that set an ideal experimental scenario, delineating the upper limit of similar knowledge forms. Conversely, the WoW dataset, with knowledge retrieved from Wikipedia or constructed via OpenIE, mirrors more closely a real-world scenario. While there are indeed other available datasets like CMU_DoG, they do not extend beyond the characteristics intrinsic to our chosen datasets. Consequently, we believe our selection of these three datasets for the study is justifiable and robust.
>
> - In response to the concern about under-representation of text and graphs structures, it should be noted that our study is predicated on employing the most straightforward knowledge augmentation methods for validation, as its primary goal is to compare graph and text sequences fairly. We acknowledge certain optimization approaches specific to each knowledge type—like graph-based reasoning and text-based retrieval—are not explicitly discussed. Although these methods are bespoke solutions well-suited to their respective knowledge types and may influence specific results, they do not impede our capacity to draw relevant general conclusions.
>
> - In response to the critique about dated dialog evaluation metrics, we acknowledge that concern. As stated in Section 4.1, we have indeed noted the limitations of automated metrics and hence incorporated human evaluation into our methodology for a more comprehensive understanding. Our analysis and conclusions incorporate both types of evaluation.
>
> - In response to the comment on results is dataset dependent, it is essential to highlight that the datasets we employed, as mentioned in Answer 1 and Section 4.1, are representative and encompass diverse aspects. The distinct characteristics of these datasets naturally yield varied conclusions. Consequently, our findings are not prescriptive, suggesting one approach over another, but rather they offer optimization methods tailored to individual datasets. Thus, our focus is on enhancing text quality, retrieving fine-grained knowledge, among other improvements, rather than asserting the superiority of one method over the other.
>
> - In responding to your observation about potential bias in factual consistency results due to text-based metrics, we appreciate the feedback. As you've duly noted, we have highlighted this limitation in our work. We plan to innovate knowledge-independent factual consistency evaluations in future studies to address this concern. Nonetheless, while this caveat is important, it does not invalidate our current findings or serve as a sufficient basis to dismiss this study.
>
> - Responding to your query about graph embedding methods and their interplay with response generation systems; indeed, a variety of such methods exist. However, these mostly focus on modeling the entire graph and do not readily adapt to knowledge-grounded conversations. The advent of pre-trained language models has led to the use of knowledge graphs as sequential text inputs becoming predominant in knowledge-based text generation. We conducted experiments using typical graph encoders (refer to Appendix C.8) and determined that it is not equitable to compare graph embeddings with large-scale pre-trained language models. Consequently, we opted for the serialized knowledge graph approach for comparison in our study.
>
> - In response to the comment about the absence of studying the role of prompting, while it's recognized that different prompts could potentially influence the generated responses, it is important to note that prompt engineering is highly individual and somewhat constrained. The central aim of our study is to compare knowledge across different forms, thus enabling us to set aside such engineered optimization strategies. Instead, we adopt a naive approach that promotes drawing broad generalizations without being overly influenced by the specificities of prompting.
>
> - We appreciate your observation regarding the omission of recent surveys in our work. We will diligently review these documents and incorporate necessary references to enhance the depth and relevance of our study in the revised version of the paper.

---

### Meta-Review · Area_Chair_WLNQ · 2023-09-18

**Recommendation:** 4

**Metareview:**

This work analyzes the strength and limitations of knowledge graphs (knowledge tuples) compare with knowledge sequences (unstructured textual contents) for knowledge grounded dialogue.

Contributions:
 - The analysis provides valuable insights for building knowledge-grounded conversation applications

Weakness:
 - Missing references and the paper's message is sometime hard to follow.

---

### Decision · Program_Chairs · 2023-10-07

**Decision:**

Accept-Main

**Comment:**

This work analyzes the strength and limitations of knowledge graphs (knowledge tuples) compare with knowledge sequences (unstructured textual contents) for knowledge grounded dialogue.

Contributions:
 - The analysis provides valuable insights for building knowledge-grounded conversation applications

Weakness:
 - Missing references and the paper's message is sometime hard to follow.